# Calystegines Improve the Metabolic Activity of Human Adipose Derived Stromal Stem Cells (ASCs) under Hyperglycaemic Condition through the Reduction of Oxidative/ER Stress, Inflammation, and the Promotion of the AKT/PI3K/mTOR Pathway

**DOI:** 10.3390/biom12030460

**Published:** 2022-03-16

**Authors:** Anna Kowalczuk, Nabila Bourebaba, Juliia Panchuk, Krzysztof Marycz, Lynda Bourebaba

**Affiliations:** 1National Medicines Institute, 00-725 Warsaw, Poland; 2Department of Experimental Biology, Faculty of Biology and Animal Science, Wrocław University of Environmental and Life Sciences, 50-375 Wrocław, Poland; nabila.bourebaba@upwr.edu.pl (N.B.); 121908@student.upwr.edu.pl (J.P.); krzysztof.marycz@upwr.edu.pl (K.M.); 3International Institute of Translational Medicine, 55-114 Wisznia Mała, Poland

**Keywords:** calystegines, hyperglycaemia, HuASCs, ER stress, inflammation, mTOR

## Abstract

Hyperglycaemia and its resulting glucotoxicity are among the most prominent hallmarks of diabetes mellitus (DM) development. Persistent hyperglycaemia further leads to oxidative stress via mitochondrial dysfunction and subsequent ER stress onset, while associated hyperlipidaemia triggers the adipose tissue to secrete pro-inflammatory cytokines. In this study, the effect of calystegines has been investigated in an experimental model of hyperglycaemia induced on human ASCs cells. Different cellular pathways including apoptosis, oxidative and ER stress, inflammation as well as Pi3K/AKT/mTOR metabolic-associated axis have been evaluated by means on RT-qPCR, western blot, and flow cytometry techniques. Treatment of HuASCs cells with calystegines strongly promoted the hyperglycaemic cells survival and significantly diminished oxidative stress, mitochondrial dynamics failure and ER stress, while improving the endogenous cellular antioxidant defenses. Interestingly, nortropane alkaloids efficiently prevented the hyperglycaemia-mediated inflammatory response, as evidenced by the regulation of the pro- and anti-inflammatory response in HuASCs cells. Finally, we evidenced that calystegines may exert their protective effect on HuASCs cells metabolic functions through the restoration of the defective PI3K/AKT/mTOR pathway. Overall, the present investigation demonstrated that calystegines possess important abilities to protect HuASCs against hyperglycaemia-induced cellular dysfunction, and it evidenced that the observed effects are associated to the promotion of PI3K/AKT/mTOR pathway.

## 1. Introduction

Diabetes mellitus (DM) nowadays represents a colossal challenge for research in medicine and therapeutics as its prevalence is constantly growing at alarming proportions worldwide [1]. This condition essentially refers to a cluster of metabolic affections related to chronic hyperglycaemia resulting from defects in insulin secretion, insulin action, or both, and that lead to the establishment of metabolic abnormalities in carbohydrates, lipids and proteins—all of which are regulated by the anabolic activities of insulin [2]. Usually derived from chronic overnutrition that results in a constant amount of elevated blood glucose that is harmful to the macrovascular and the microvascular systems, hyperglycaemia and its resulting glucotoxicity are among the most prominent hallmarks of DM development [3]. Persistent hyperglycaemia undeniably leads to oxidative stress via mitochondrial dysfunction and subsequent increased reactive oxygen species (ROS) production, while associated hyperlipidaemia triggers the adipose tissue to secrete pro-inflammatory cytokines. The resulting oxidative stress and low-grade inflammation have been identified as major molecular players in the pathogenic progression of T2DM and its complications [4]. Hyperglycaemia-mediated cell and tissue dysfunction can occur secondary to a concomitant increased ER stress. This adaptative response has been largely correlated with ROS overgeneration as oxidative stress can also induce ER stress [5]. Under physiological stat, ER stress is considered as a protective cellular process that is known as the unfolded protein response (UPR), which strives to restore the natural cellular metabolic balance in the course of hyperglycaemia. However, if the hyperglycaemia sustains and the cellular homeostasis remains disrupted, an ER stress response ultimately initiates cell death stimuli, leading to ER stress-induced apoptosis [6].

Increased visceral adiposity combined with existing hyperglycaemia are known to alter the functions and the secretory profiles of adipose tissue-resident cells. As a consequence, pro-inflammatory adipokines including tumour necrosis factor α (TNFα), interleukin-6 (IL-6) and C-reactive protein (CRP) are overproduced, while others with anti-inflammatory or insulin-sensitizing properties, such as adiponectin, are decreased [7]. Recent studies suggested that adipose-derived stromal stem cells (ASC), which are fibroblast-like stem cells characterised by self-renew and multipotency abilities in addition to immunomodulatory and anti-inflammatory properties under physiological conditions, undergo many metabolic changes in the course of T2DM. Indeed, ASCs isolated from diabetic patients exhibited strong mitochondrial dysfunction and senescence phenotype, significant reduced cell proliferation, impaired glucose homeostasis and insulin sensitivity due to excessive oxidative stress [8,9,10]. Furthermore, it has been proposed that in T2DM, the adipose tissue microenvironment displays the highest levels of oxidative stress, glycemia and hypoxia, all of which may significantly contribute to a disruption of the ASC niche, which could therefore reverse their immunomodulatory trend and trigger the overproduction of pro-inflammatory cytokines to the detriment of anti-inflammatory molecules [11]. The PI3K/AKT/mTOR pathway has recently emerged as a key regulator of stem cell properties and functions that essentially regulates cellular proliferation and survival, gene expression, epigenetic configuration and metabolic activities, and whose alteration may exert dramatic consequences on the properties and functions of these cells [12].

Multiple studies have demonstrated that the PI3K/AKT/mTOR pathway plays a critical role in maintaining glucose homeostasis. The major cellular and molecular events that it underlines are key components for cellular response to extracellular stimuli, such as insulin and insulin-like growth factor-1 (IGF-1) [13]. When elicited, insulin induces its receptor phosphorylation and subsequent insulin receptor substrate phosphorylation at tyrosine residue, which consequently activates phosphatidylinositol-3 kinase (PI3K) and downstream conversion of phosphatidylinositol-4,5-bisphosphate (PIP2) into phosphatidylinositol-3,4,5-trisphosphate (PIP3). Protein kinase B (AKT) is afterwards translocated to the membrane and further recruits mTOR protein complex, which exerts plenty of metabolic effects, including nutrient uptake and glucose homeostasis control [14]. Uncoupling of downstream insulin signalling at the PI3k-Akt-mTOR axis in various cell types including hepatocytes and adipocytes has been directly implicated in the development of insulin resistance and type 2 diabetes. Moreover, recent lines of evidence indicate that chronic hyperglycaemia-induced uncoupling of post-receptor insulin signalling at PI3k-Akt in endothelial cells leads to vascular complications in diabetes [15]. In view of the emerging importance of the Pi3K/AKT/mTOR pathway in the regulation and the maintenance of cellular metabolic homeostasis, it is reasonable to consider it as a new target for the development of new anti-hyperglycaemic drugs.

Phytochemicals have long been regarded as a valuable source of treatment for human health issues. As a matter of fact, phytotherapy has been shown to be effective in the management of diabetes in several experimental and clinical trials [16]. Plant derivative anti-diabetic effects are mediated through a variety of molecular pathways, including decreased intestinal glucose absorption, inhibition of liver gluconeogenesis, increased glucose uptake by tissues, improved insulin secretion from β-cells, and/or increased pancreatic tissue regeneration [17].

Azasugars, also designated as iminosugars, have recently gained considerable interest as biological matrices for the synthesis of new therapeutic agents. The biological and the therapeutic values of iminosugars arise from their natural structure that enables an efficient intermeddling with the function of carbohydrate-processing enzymes and carbohydrate-recognizing receptors, which take place in the different cellular metabolic reactions in all living organisms [18]. Among the wide variety of discovered natural iminosugars, calystegines represent a new class of polyhydroxylated alkaloids carrying an azapyranose ring with a tertiary hydroxyl group at the bicyclic bridgehead of their structure, and initially isolated from *Atropa belladonna* (Solanaceae) and *Convolvulus arvensis* (Convolvulaceae), but later found in other species such as *Datura*, *Morus* or *Hyoscyamus* [19,20]. Previous studies already reported on the potent inhibitory effects of calystegines toward various intestinal and liver α- and β-glucosidase specific disaccharidases involved in mammalian digestion, including sucrase, maltase, isomaltase, etc., thus demonstrating the potential of calystegines in carbohydrate adsorption regulation and overall carbohydrates metabolism homeostasis [21,22]. Therewith, our previous published data highlighted the potent ability of total calystegines to rescue adipose derived stromal stem cells (ASCs) isolated from horses suffering from equine metabolic syndrome (EMS) in terms of cell survival through apoptosis mitigation, mitochondrial dynamics improvement and insulin sensitivity restoration [23]. In the light of these observations, the present investigation was based on a verification of the hypothesis according to which the application of a calystegine mixture could improve the metabolic activity of human ASCs cells cultured in a hyperglycaemic milieu through essentially the reduction of oxidative and ER stress and inflammation, emphasizing particularly the potential changes in the PI3K/AKT/mTOR signalling pathways.

## 2. Materials and Methods

### 2.1. Chemicals

All reagents and chemicals used in the experimental section of the present study were purchased from Sigma Aldrich (Poznań, Poland). The cell culture media and the reagents were bought from BioWest (VWR International, Gdańsk, Poland). The other reagents and chemical specificities, if any, have been described in the manuscript.

### 2.2. Isolation and Characterisation of Calystegines

The *Hyoscyamus albus* plant was identified and collected in Algeria in a semi-arid climate and low rainfall. After cleaning, the plant seeds were dried in a dark ventilated room and subsequently crushed using an electric grinder to obtain a fine powder of 125 µm granulometry.

Calystegines were isolated as previously described [24]. Briefly, air-dried and ground plant material was extracted three times by maceration in water/methanol (50/50, *v*/*v*) for 24 h. After centrifugation, the supernatant was evaporated to dryness and applied to a cation-exchange column (Amberlite IR 120B, H+ form), followed by an anion-exchange column (Dowex 1 × 2, Cl-form) and eluted with water to obtain a purified plant extract containing total calystegines. The extract composition has been evaluated using gas chromatography–mass spectrometry (GC-MS) as trimethylsilyl (TMS) ether derivatives (Figure 1), which demonstrated the presence of: three A group calystegines (A3: 54.9 µg/g DW, A5: 45.91 µg/g DW and A5 glycoside: 197.76 µg/gDW); calystegine N1: 180.22 µg/g DW; as well as three B group calystegines (B1: 91.25 µg/gDW, B2: 87.14 µg/gDW and B4: 212.54 µg/g DW)—calystegine B4 being the majority isolated nortropane alkaloid [23].

### 2.3. Human Adipose Derived Stromal Stem Cells (HuASCs) Isolation and Culture Conditions

Human adipose derived stromal stem cells were isolated from subcutaneous adipose tissue (AT) biopsies collected from healthy donors (both sexes of people aged from 38 to 56; BMI = 15.45 ± 2.21; n = 6), during a total hip arthroplasty surgical procedure. Forthwith after tissue collection, samples were transferred to tissue transport medium consisting of Hank’s balanced salt solution (HBSS) supplemented with 1% antimycotic antibiotic solution (penicillin/streptomycin/amphotericin B, P/S/A). Immediately after preparation, pieces of adipose tissue were washed with phosphate buffered saline (PBS) supplemented with 1% P/S, mechanically diced and treated with a 1 mg/mL solution of type I collagenase at 37 °C for 40 min. After incubation, the digested tissue was centrifuged (1200× *g* for 10 min) to obtain a cell pellet. After that, the supernatant was discarded and the cell pellet was resuspended in PBS and centrifuged again (300× *g* for 4 min). After the second centrifugation, the cells were seeded in T surface 25 cm^2^ polystyrene culture flasks and cultured in Dulbecco’s Modified Eagle Medium (DMEM) with nutrient F-12 Ham supplemented with 10% of foetal bovine serum (FBS), 1% P/S/A and 0.5% gentamicin, upon reaching approximately 85% confluence. Cells were then carefully recovered using TrypLE Express solution (Life Technologies, Carlsbad, CA, USA) and re-seeded into a T surface 75 cm^2^ flask. The medium was refreshed every 2–3 days. Before the experiment, the cells were passaged three times.

### 2.4. Biocompatibility of Hyoscyamus albus Total Calystegines

The cell viability and the proliferative rate were determined using a resazurin-based assay kit (TOX-8) to estimate the cellular metabolic activity and a BrdU cell proliferation ELISA (Abcam, Cambridge, UK), respectively. To assess cells’ metabolic status, thrice passaged HuASCs were seeded onto 96-well plates at a density of 2 × 10^3^ cells per well, and treated with various calystegine concentrations (50 to 300 µg/mL). After 24 h, 48 h and 72 h of cultivation, the culture media were replaced with a 10% resazurin dye solution prepared in a fresh complete culture medium. After a 2 h incubation at 37 °C in a CO_2_ incubator, absorbances were measured spectrophotometrically (Epoch, Biotek, Bad Friedrichshall, Germany) at 600 nm and 690 nm as reference wavelengths. Similarly, the cell proliferation rate was evaluated by labelling the incorporated BrdU in treated cells with the anti-BrdU antibody for 1 h. Afterwards, a horseradish peroxidase-conjugated goat anti-mouse antibody was added to each well and incubated for 30 min at room temperature. After washing and incubation with 3,3′,5,5′-tetramethylbenzidine (TMB) chromogenic substrate, the reaction product was quantified spectrophotometrically (Epoch, Biotek, Bad Friedrichshall, Germany) at a wavelength of 450 nm.

### 2.5. In Vitro Hyperglycemia Induction and Related Treatments

The experimental model of hyperglycemia (HG) was established by exposing HuASCs cells to high levels of glucose as follows. In brief, human adipose derived stromal stem cells were seeded on culturing plates and kept overnight under standard culturing conditions for cellular attachment. The cells were then pretreated with two optimal calystegine concentrations, i.e., 125 and 250 μg/mL for 24 h. All experimental groups were subsequently exposed to a high D-glucose concentration (30 mM) in a serum-free medium for another 24 h after overnight serum starvation. A group of untreated healthy human adipose derived stromal stem cells (HuASCs-HE) was included and cultured under the same experimental conditions to serve as a control group.

### 2.6. Cell Survival Evaluation Following Hyperglycemia Induction

HuASCs cell survival under hyperglycaemic conditions has been evaluated using a resazurin-based assay kit (TOX-8), which estimates the cellular metabolic activity of living cells. Briefly, HuASCs were seeded on 96-well plates at a density of 2 × 10^3^ cells per well and pretreated with two optimal concentrations of calystegines (125 and 250 µg/mL). After 24 h, hyperglycemia was induced by supplementing culture media with 30 mM D-glucose after overnight starvation. The culture media were replaced with a 10% resazurin dye solution prepared in fresh complete culture medium following 24 h exposure to glucose. After a 2 h incubation at 37 °C in a CO_2_ incubator, absorbance levels were measured spectrophotometrically (Epoch, Biotek, Bad Friedrichshall, Germany) at 600 nm and 690 nm as reference wavelengths.

### 2.7. Bromodeoxyuridine (BrdU) Incorporation Assay

To determine the cell proliferative rate using the BrdU cell proliferation ELISA kit, cells were seeded on 96-well plates at a density of 6 × 10^3^ cells/well in 100 μL of culture medium. The procedure was performed according to the supplier’s instructions. After each related treatment with calystegines and a high D-glucose concentration, 20 μL of the diluted BrdU reagent was added directly to the culture media and incubated overnight at 37⁰C under standard growth conditions. The cells were then fixed, permeabilized and incubated with a monoclonal anti-BrdU antibody detector. After 1 h incubation, goat anti-mouse antibodies conjugated with horseradish peroxidase were added and incubated for 30 min at room temperature. After washing with HBSS, and incubation with 3,3′,5,5′-tetramethylbenzidine (TMB) chromogenic substrates, the stained reaction product was quantified by spectrophotometric measurement (Epoch, Biotek, Bad Friedrichshall, Germany) at a wavelength of 450 nm.

### 2.8. Cell Apoptosis Analysis Using Flow Cytometry

The apoptosis rate was assessed using the Muse^®^Annexin V & Dead Cell Assay kit. All cultured HuASCs that had been treated with the calystegines, healthy cells group (HuASCs-HE) and hyperglycaemic cells group (HuASCs-HG) were collected, washed with HBSS solution containing 1% FBS and labeled with the Annexin V & Dead Cell ready to use reagent for 20 min at room temperature. The percentage of live, early apoptotic, late apoptotic and dead cells was then determined following the manufacturer’s instructions, using a Muse Cell Analyzer (Merck Millipore, Darmstadt, Germany) as follows: (i) non-apoptotic cells, not undergoing detectable apoptosis, Annexin V (−) and 7-AAD (−); (ii) early apoptotic cells, Annexin V (+) and 7-AAD (−); (iii) late apoptotic cells, Annexin V (+) and 7-AAD (+); and (iv) cells that have died through non-apoptotic pathway, Annexin V (−) and 7-AAD (+).

### 2.9. Oxidative Stress Measurement

Intracellular reactive oxygen species (ROS) were determined using the Muse^®^ Oxidative Stress Kit based on dihydroethidium (DHE). The analysis distinguished between two cell populations: live ROS cells (−) and ROS (+) cells exhibiting high content of ROS. All procedures were performed according to the protocols provided by the supplier. Cells were collected using Trypsin–EDTA 1× in PBS without calcium, magnesium and phenol red following the above-described treatments, washed with HBSS and mixed with the Muse Oxidative Stress Reagent working solution prepared in 1× Assay Buffer. After 30 min incubation at 37 °C, samples were proceeded using a Muse™ Cell Analyzer (Merck, Darmstadt, Germany).

### 2.10. Analysis of Mitochondrial Transmembrane Potential

The mitochondrial depolarization of the inner membrane (ΔΨ) was evaluated using Muse™ MitoPotential Kit from Merck Millipore according to supplier’s protocol. HuASCs cells were seeded onto 6-well plastic plates in the complete DMEM and incubated at 37 °C under the same previously detailed experimental conditions. After incubation, the cells were collected using Trypsin–EDTA 1× in PBS, washed with HBSS, mixed with the MitoPotential working solution prepared in 1× Assay Buffer (1:1000), and incubated for 20 min at 37 °C. After the incubation time, the Muse MitoPotential 7-AAD reagent was added to the cell suspension and further incubated for 5 min at room temperature in the dark. Cells were analysed using the Muse™ Cell Analyzer equipped with Muse™ Software.

### 2.11. Mitochondrial Network Staining

Mitochondrial morphology changes were observed using a confocal microscope (Observer Z1 Confocal Spinning Disc V.2 Zeiss with live imaging chamber). In brief, the HuASCs untreated and treated cells as described above seeded onto glass coverslips were incubated in the cell culture incubator for 30 min in the presence of the MitoRed (Sigma Aldrich, Poznań, Poland) fluorescent dye (1:1000 in culture medium) for mitochondria staining. Afterwards, the excess MitoRed was washed out using HBSS and cells were fixed with 4% paraformaldehyde (PFA) at room temperature for 45 min. The nuclei were counterstained used the diamidino-2-phenylindole (DAPI) contained in the used ProLongTM Diamond Antifade Mountant with DAPI (InvitrogenTM, Warszawa, Poland). All images were captured using a Canon PowerShot camera. Acquired photomicrographs were analysed and merged using ImageJ software (Bethesda, MD, USA).

### 2.12. Inflammatory Cytokines Measurement Using ELISA Assays

The anti-inflammatory effect of the calystegines was assessed from collected conditioned media using the TNF-α (Human TNF-α (Quantikine ELISA Kit, R&D Systems, Minneapolis, MN, USA) and IL-1β (R&D Systems, Minneapolis, MN, USA) ELISA kits following the manufacturer’s instructions. The absorbance was measured with a 96-well microplate reader (Epoch, Biotek, Bad Friedrichshall, Germany) at 450 nm.

### 2.13. Endogenous Antioxidant Enzyme Activity Assays

Superoxide dismutase enzyme activity was measured using the SOD assay kit-WST (ScienCell Research Laboratories, San Diego, USA), and catalase enzyme activity was measured using the catalase (CAT) colorimetric assay Kit (MyBioSource, San Diego, USA). Briefly, 1 × 10^6^ cells were incubated with calystegines extract for 24 h and, after that, hyperglycemia was induced with 30 mM D-glucose for 24 h. the cells were then washed with cold-HBSS and suspended in a cold-lysis assay buffer. The supernatants containing proteins were used for the determination of SOD and CAT enzyme activities. These were added to each well in microtiter plates with the appropriate working solutions (according to the manufacturer’s instructions). The colour changes were measured at 438 nm for SOD and 240 nm for CAT, using a microplate reader (Epoch, Biotek, Bad Friedrichshall, Germany). the results were expressed as a percentage of SOD and nmol of decomposed H_2_O_2_/min/mg protein for CAT.

### 2.14. Reverse Transcription Quantitative Real-Time PCR Analysis

The HuASCs cells were seeded onto the 6-well plastic plates and incubated with calystegines prior to hyperglycemia induction. After the incubation time, the cells were suspended in TRIzol reagent for RNA isolation following the manufacturer’s instructions. The quantity and the purity of the isolated RNA was assessed using a spectrophotometer at 260 nm wavelength (Epoch, Biotek, Bad Friedrichshall, Germany). A total of 150 ng RNA was used for cDNA synthesis using a Tetro cDNA Strand cDNA Synthesis Kit (Bioline, London, UK) based on oligo (dT) primers in a T100 Thermal Cycler (BioRad, Hercules, CA, USA). The detection of the target gene expression was performed using a SensiFAST SYBR Green Fluorescein Kit (Bioline, London, UK) in a CFX Connect™ Real-Time PCR Detection System (BioRad, Hercules, CA, USA). The relative expression of genes associated with apoptosis, oxidative stress, mitochondrial dysfunction inflammation and ER stress (Table 1) has been normalized to GAPDH (glyceraldehydes-3-phosphate, housekeeping gene) expression and calculated using the 2^−ΔΔcq^ method.

### 2.15. Western Immunoblot Analysis

HuASCs cells were collected from each experimental culture dish and homogenized in a lysis buffer (Tris at 50 mmol/L pH 7.4, NaCl at 150 mmol/L, SDS 0.1%, sodium deoxycholate 0.5%, protease cocktail, 1 mmol/L PMSF, 10 mmol/L of sodium ascorbate, 1% Triton X-100, 10 mmol/L sodium azide and Trolox at 5 mmol/L) in the presence of a phosphatase and protease inhibitor cocktail on ice in order to analyse the protein profiles. After centrifugation for 20 min at 4 °C and 6000× *g*, the sample supernatants were transferred to fresh tubes in order to remove the insoluble materials and kept at −80 °C until further use. The protein concentration was determined using the Pierce™ bicinchoninic acid (BCA) protein assay kit (InvitrogenTM, Warszawa, Poland), and cell lysates were initially diluted with a 4 × Laemmli loading buffer (Bio-Rad, Warszawa, Poland) and subsequently denatured at 75 °C for 10 min. the samples were subjected to SDS–polyacrylamide gel electrophoresis in a Tris/glycine/SDS buffer at 100 V using Mini PROTEAN Tetra Vertical Electrophoresis Cell (Bio-Rad, Warszawa, Poland) for 90 min and transferred to the polyvinylidene difluoride (PVDF) membranes (Bio-Rad, Warszawa, Poland) with a Mini Trans-Blot^®^Cell (Bio-Rad, Warszawa, Poland) transfer apparatus in a Tris/glycine buffer/methanol at 100 V, 250 mA at 4 °C for 45 min. Obtained membranes were blocked in a 5% non-fat milk solution in TBST. The proteins were detected with primary antibodies (Table 2) by incubation overnight at 4 °C and HRP-conjugated secondary antibodies (dilution 1:2500 in TBST, 1 h incubation at room temperature). Chemiluminescent signals were screened and quantified with Image Lab Software (Bio-Rad, Warszawa, Poland), using the ChemiDoc MP Imaging System (Bio-Rad, Warszawa, Poland).

### 2.16. Statistics

All statistical analyses were performed using GraphPad Prism 8.0 (San Diego, CA, USA) with a one-way analysis of variance (ANOVA) followed by Bonferroni’s post-hoc multiple comparison test, as indicated. Asterisk (*) and Hash (#) signs indicate statistical significance in the HG-induced groups versus the healthy control or in the HG-induced control versus the calystegine-treated groups, respectively. All *p* values lower than 0.05 (*p* < 0.05) are summarized with one asterisk/hash (*/#), those at *p*< 0.01 use two asterisks/hashes (**/##) and those at *p* < 0.001 have three asterisks/hashes (***/###).

## 3. Results

### 3.1. Calystegines Are Cytocompatible with HuASCs

The concentration–dependent cytocompatibility of calystegines isolated from *H. albus* seeds has been evaluated in terms of cell viability and proliferation. Incubation of HuASCs cells in the presence of various concentrations of calystegines during 24 h, 48 h and 72 h did not induce any increased cytotoxicity as compared to unstimulated controls, up to a maximal concentration of 300 µg/mL at all incubation time points (Figure 2a). What is more, the alkaloidal extract application resulted in a significantly increased number of living cells (Figure 2a) when compared to untreated cells, starting from a concentration of 200 µg/mL (*p* < 0.05). These observations have been further confirmed by the assessment of BrdU-labeled cells, where a stimulation of the HuASCs proliferation rate has been observed in the groups treated with concentrations ranging from 200 to 300 µg/mL of calystegines (Figure 2b).

### 3.2. Calystegines Reduce HuASCs Apoptosis under Hyperglycaemic (HG) Condition

The potential protective effect of calystegines on HG-induced HuASCs cell apoptosis has been evaluated using the Tox8, BrdU and MUSE Annexin V & Dead Cell assays, as well as gene expression analysis of pro-apoptotic factors following each related treatment. Obtained data revealed an obvious reduction in the number of viable and proliferative cells upon exposure to a high concentration of D-glucose (Figure 3a,b), when compared to untreated cells (*p* < 0.01). Moreover, the HG-treated group exhibited a significantly higher amount of apoptotic and dead cells (Figure 3d). The same group of cells further displayed significant upregulated pro-apoptotic gene expression including *p53*, *p21* and *Bax* (*p* < 0.001) and simultaneous suppression of *Bcl-2* survival gene expression (*p* < *0.01*). Similarly, hyperglycaemic cells were characterised by increased caspases transcript levels (*Casp3*, *Casp6*, *Casp7*, *Casp8* and *Casp9*) as compared to unchallenged HuASCs cells (*p* < 0.001). The pre-treatment of HuASCs cells with calystegines prior to HG induction showed a protective effect of the alkaloids against apoptosis. Indeed, the calystegines at the two tested concentrations significantly prevented cell death and enhanced proliferation as evidenced by the increase in newly synthetized DNA (Figure 3b), when compared to HG-untreated cells (*p* < 0.01). Moreover, the nortropane alkaloids enabled a marked regulation of apoptosis related effectors through the downregulation of *p53*, *p21* and *Bax* as well as caspases transcripts (Figure 3e,f). It should also be noted that the alkaloid extract prevented *Bcl-2* cell survival gene expression suppression at the highest concentration, i.e., 250 µg/mL as compared to both the healthy and the HG control groups.

### 3.3. Calystegines Exhibit Antioxidant Effect on Hyperglycemia Induced Oxidative Stress in HuASCs Cells

Calystegines are polyhydroxylated alkaloids that may represent potentially potent antioxidant substances, which could ameliorate the characteristic oxidative stress of cells affected by hyperglycemia and glucotoxicity in the course of severe metabolic disorders. Therefore, we undertook to evaluate the intracellular ROS levels as well as the status of endogenous antioxidant enzymes in hyperglycaemic HuASCs cells after treatment with calystegines. As illustrated in Figure 4b, HuASCs cells were characterised by a significant increase in intracellular ROS namely superoxide radicals when cultured in the presence of a high concentration of D-glucose, indicating the onset of oxidative stress in response to glucotoxicity (*p* < 0.001). As a consequence, RT-qPCR data (Figure 4c) showed that hyperglycaemic untreated ASCs cells exhibited dysregulated endogenous antioxidant defenses as evidenced by the significant overexpression of *Sod1*, *Sod2* and *Cat* transcripts when compared to healthy cells (*p* < 0.001), suggesting the establishment of a compensatory mechanism aimed at overcoming the breakdown of endogenous antioxidant enzymes; this correlates with the observed remarkable drop in the intrinsic activities of the two SOD and CAT enzymes in the HG-nontreated HuASCs cells (Figure 4d). The pre-treatment of HuASCs cells with two different concentrations of calystegines (125 and 250 µg/mL) prevented the overproduction of ROS in cells challenged with D-glucose, as demonstrated by the reduced number of ROS-positive cells (Figure 4b). Moreover, the alkaloids considerably improved the cellular antioxidant enzyme functions, mainly by the stimulation of SOD and CAT activities and thus by regulating the expression of *Sod1*, *Sod2*, *Cat* and *GPx* at the mRNA level in opposition to untreated hyperglycaemic HuASCs cells (Figure 4c,d).

### 3.4. Calystegines Ameliorate HuASCs’ Mitochondrial Dynamics Defects under Hyperglycaemic Condition

As cell powerhouses, mitochondria are known to primarily participate in overall cellular ROS overproduction, as well as apoptosis event engagement, when they undergo metabolic disruption. In this regard, the influence of calystegines on hyperglycaemia induced mitochondrial failure has been investigated in order to shed light on the crosstalk between the observed antiapoptotic and antioxidant effects of tested alkaloids and HuASCs’ mitochondria functions. The obtained data demonstrated that exposure of HuASCs cells to high toxic levels of D-glucose triggered mitochondrial activity collapse, as underlined by the remarkable increase in total live and dead cell amounts with depolarized mitochondrial membrane potential (Figure 5b) in comparison to cells cultured under physiological D-glucose concentration (*p* < 0.01; *p* < 0.001). Moreover, a considerable failure in mitochondrial dynamics has been evidenced. Indeed, hyperglycemia led to the significant downregulation of mitochondrial fusion related actors, including *MFN-1*, *MFN-2* and *OPA-1* (Figure 5c), while inducing the expression of pro-fission markers namely *DRP-1* and *Fis-1* (*p* < 0.001) and confirming that overaccumulation of glucose leads to impaired mitochondria characterised by enhanced division and fission, which consequently culminate to excessive mitochondrial reactive oxygen species production. The application of calystegines prior to hyperglycemia induction sensibly improved the mitochondrial machinery; hence, the nortropane alkaloids enhanced mitochondrial activity and cellular health as demonstrated by the sharp decrease in cells with depolarized mitochondria (Figure 5b) in contrast to HG untreated HuASCs cells. Therewith, cells pretreated with calystegines displayed mild mitochondrial dynamic perturbations, as evidenced by the positive regulation of the fusion/fission balance and the consequential suppression of excessive *DRP-1* and *Fis-1* transcript expression, as well as the maintenance of *MFN-1* and *OPA-1* transcription (Figure 5c). These observations are supported by the mitochondrial morphology evaluation results, where calystegine treated cells demonstrated a more homogeneous dense mitochondrial network, with higher florescence intensity that evoked greater mitochondrial activity when compared to the HG untreated cells that displayed a more fragmented mitochondrial network (Figure 5d). These findings thus suggest that calystegines reduce oxidative stress and apoptosis through the modulation of imbalanced mitochondrial dynamics.

### 3.5. Calystegines Mitigate ER Stress Onset in HuASCs Cultured under Hyperglycaemic Milieu

Persistent hyperglycemia negatively affects endoplasmic reticulum (ER) homeostasis, which consequently participates in oxidative stress exacerbation and ultimately cell apoptosis. As such, total calystegines isolated from *H. albus* seeds were tested for their impact on ER stress related factors expression using RT-qPCR technique. As reported in Figure 6, HuASCs cells cultured in the presence of a high D-glucose concentration were prone to excessive ER stress, with a significant elevation in *ATF6*, *PERK*, *CHOP* and *IRE1-α* fold gene expression in relation to healthy untreated HuASCs cells (*p* < 0.001). The active calystegine extract applied as a pre-treatment to the HuASCs enabled strong modulation of the overexpression of *ATF6*, *PERK* and *IRE1-α* ER sensors as well as *CHOP* pro-apoptotic effector protein in contrast to hyperglycaemic untreated cells (Figure 6), suggesting that calystegines counteract a hyperglycemia-induced loss of homeostasis in the ER.

### 3.6. Calystegines Regulate the Inflammatory Profile of HuASCs Cells Elicited with a High D-Glucose Level

Adipose tissue is considered as being very sensitive to chronic hyperglycaemia that triggers the long-term inflammation and overproduction of reactive oxygen species, which further exacerbates inflammatory processes in residing cells including ASCs. Along these lines, inflammation-related cytokine expression at both the mRNA and the protein levels were measured by quantitative RT-PCR and ELISA assays in hyperglycaemic HuASCs treated with a total calystegine extract. Increased levels of different pro-inflammatory cytokine genes (Figure 7a), such as interleukin-1β (*IL-1β*), interleukin-4 and -6 (*IL-4*/*6*) and tumour necrosis factor-α (*TNF-α*) have been found in HuASCs challenged with a high concentration of D-glucose (*p* < 0.001), while a profound downregulation of the main anti-inflammatory factor transcripts including *IL-10* and *TGF-β* (Figure 7b) has been observed for the same group of cells (*p* < 0.001). Similarly, ELISA quantification results demonstrated a significant rise in the released protein concentrations of both IL-1β and TNF-α master pro-inflammatory cytokines (Figure 7c) when culturing HuASCs cells in a D-glucose-rich culture medium, in opposition to cells cultured in a normoglycemic concentration (*p* < 0.001; *p* < *0.01*). Preconditioning HuASCs cells with calystegines prior to HG induction engendered a notable regulation of the pro-inflammatory cytokine’s secretion profile, as highlighted by the observed downregulation of *IL-1β* and *TNF-α* at both the mRNA and the protein levels (Figure 7a,c). Therewith, alike trends were recorded in the case of the *IL-10* and the *TGF-β* anti-inflammatory cytokines, whose relative expression of their encoding genes has increased following treatment with calystegines, hinting at the potent regulatory effect of calystegines towards the impaired immunomodulatory properties of HuASCs cells affected by hyperglycaemia.

### 3.7. Calystegines Restore the Impaired Akt/Pi3K/mTOR Signaling Pathway in Hyperglycaemic HuASCs Cells

The phosphatidylinositol 3-kinase/protein kinase B/mammalian target of rapamycin (PI3K/AKT/mTOR) signaling pathway that plays a fundamental role in the metabolic and mitogenic actions of insulin and insulin-like growth factor 1 has been reported to be sensibly altered in the course of type 2 diabetes and the underlying hyperglycemia. To this effect, the impact of calystegines applied pre-treatment on the HG-triggered defective PI3K/AKT/mTOR pathway has been assessed in HuASCs cells using RT-qPCR and western blot techniques. As outlined in Figure 8a,b, the high glucose concentration induced significant changes in the expression of PI3K, AKT and mTOR factors when compared to healthy normoglycemic cells (*p* < 0.001). As a matter of fact, the three markers appeared to be visibly downregulated following HG induction at both the mRNA and the protein levels. Interestingly, the experimental group of HuASCs cells that was preconditioned with a total calystegine extract displayed an improved kinases/mTOR pathway as evidenced by the increase in *PI3K*, *AKT* and *mTOR* transcript and protein expression levels, in opposition to the HG untreated cells (*p* < 0.001).

## 4. Discussion

The incidence of hyperglycemia is steadily growing all over the world due to increased obesity, a sedentary life style and the aging of populations all over the world. Hyperglycemia becomes a critical factor contributing to obesity development and leading to adipose tissue metabolism deterioration [25,26]. This subsequently strongly impairs the cytophysiological features of residing adipose tissue stem progenitor cells (ASCs) that under hyperglycaemic conditions are losing their unique properties [27].

To this extent, we undertook in the present study to investigate the impact of calystegine pre-treatment on human ASCs cells cultured under hyperglycaemic conditions in terms of cell survival, inflammation, oxidative and ER stress, as well as mitochondrial dynamics, and we verified a potential pathway involved in the regulation of ASCs metabolic activity. Our outcomes demonstrated that calystegines under a hyperglycaemic condition, potently ameliorate overall HuASCs metabolic activity by promoting their survival and proliferative activity as well as protecting them against apoptosis. Calystegines prevented HuASCs cellular death by averting the elevation in relative expression of *p21*, *p53* and *BAX* transcripts, while substantially maintaining *BCL-2* pro-survival gene expression. Moreover, our obtained data showed that the nortropane alkaloids effectively hampered the overexpression of a set of caspases—including Caspase-3, -9, -7, -8 and -6—that are critically involved in hyperglycemia-dependent apoptosis, oxidative and ER stress, as hyperglycemia has been shown to critically modulate cellular caspase inhibitors, including the X-linked inhibitor of apoptosis (XIAP) protein [28]. These observations thus suggest the protective effect of calystegines on HuASCs cells in an unfavorable hyperglycaemic microenvironment. Indeed, previously published data similarly reported on the anti-apoptotic effect of calystegines in ASCs isolated from horses suffering from equine metabolic syndrome (EMS), where the treatment of EMS ASCs cells with the nortropanes considerably improved cellular viability and proliferation through the suppression of pro-apoptotic overexpression [23]. Moreover, we further demonstrated that calystegines isolated from *Hyoscyamus albus* seeds exhibited cytoprotective effect with regard to the HepG2 human hepatocarcinoma cell line under insulin-resistant conditions induced by concomitant hyperglycemia and hyperinsulinemia via similarly modulating the same apoptotic signaling pathway [29].

A strong body of evidence indicates that excessive accumulation of nutrients, including glucose in tissues and cells inevitably results in a chronic overproduction of pro-inflammatory mediators and it contributes to ROS overproduction, which can ultimately lead to various cellular dysfunctions, such as insulin resistance and decreased insulin secretion [30]. In this study, we demonstrated that ASCs which are commonly used as a therapeutic factor in many disorders, including insulin resistance treatment [31], are losing their protective immunomodulatory nature and their anti-oxidative properties under hyperglycaemic conditions. For that reason, we were interested in whether in vitro application of calystegines would reverse that failure tendency. Our investigation revealed that HuASCs under hyperglycaemic conditions, and when pretreated with calystegines, are characterized by reduced intracellular ROS levels, enhanced *Sod1*, *Sod2*, *Cat* and *GPx* antioxidant gene expression, and potentiated endogenous SOD and CAT antioxidant enzymatic activities, referring to prominent diminished oxidative stress. The obtained data are strongly related to our previous findings that showed that calystegines mitigate oxidative stress in equine ASCs affected by metabolic syndrome and therefore increase their sensitivity to insulin [23]. Furthermore, in a previous investigation, *Bourebaba* et al. [24], evaluated the in vitro antioxidant effect of the same compounds and evidenced that calystegines possess strong radical scavenging effects toward DPPH and ABTS free radicals, while they conferred significant protection of human erythrocyte toward hemolysis mediated by peroxyl free radicals and attributed the observed effects to their polyhydroxylated unique structure. Chronic exposure to a high glucose concentration usually augments the metabolic flux in mitochondria through the hexosamine pathway, leading to increased ROS generation and lately to insulin secretion perturbation through multiple downstream mechanisms, including mitochondrial dynamic impairment [32]. Here, we found that maintenance of HuASCs cells in a hyperglycaemic milieu strongly affected their mitochondrial functions, as demonstrated by the sharp loss in transmembrane potential and the significant imbalanced mitochondrial fission and fusion machinery; and, therefore, the application of a pre-treatment with calystegines made it possible to considerably prevent mitochondria dynamic deregulation through the restoration of *MFN-1*, *MFN-2* and *OAP-1* gene expression—which are the main fusion-associated mediators—and the simultaneous suppression of fission related effector overexpression, namely *FIS-1* and *DRP-1*. Based on these findings, we can speculate that because apoptosis usually includes mitochondrial injury, production of ROS and oxidative stress, the mechanisms by which calystegines rescue HuASCs cells from apoptosis may include an improvement of mitochondrial dynamics and a resulting mitigation of oxidative stress [33].

Besides reducing apoptosis, the increased anti-oxidative protection of ASCs due to calystegine application possess a dual beneficial effect in the course of hyperglycemia, including an anti-inflammatory effect as well as ER stress alleviation—a critical inducer of obesity and insulin resistance that has been observed in human obesity and non-alcoholic fatty liver disease (NAFLD)-derived liver and/or adipose tissue [34]. Our study and the previous studies of others showed that ASCs isolated from T2D patients suffer increased ER stress, which is a potent risk factor of the development of insulin resistance [9]. Here, we demonstrated that calystegines become a potent factor in the regulation of oxidative as well as ER stress. We found that calystegines reduced the expression of *CHOP*, *PERK*, *ATF6* and *IRE1*—the critical sensors and regulators of ER stress. Our data and the previous data of others clearly showed that ER stress is recognized as a critical factor involved in obesity and the development of insulin resistance. Moreover, a recent study by *Liu and colleagues* showed that chronic low-grade ER stress in lean mice promoted hyperglycemia due to enhanced hepatic gluconeogenesis via ubiquitin-specific peptidase14 (USP14) induction [35]. The study of *Cnop* et al. [36] demonstrated that ER stress contributes to the development of steatosis and insulin resistance not only in liver cells but also in adipose tissue, leading to the modification of adipokine profile secretion. Moreover, the data presented by *Kawasaki and colleagues* clearly proved that ER stress represents one of the most important factors that are involved in obesity and obesity-related inflammation development [37]. Similar to the presented effect on human ASCs cells, calystegines have also been previously demonstrated to reduce aberrant ER stress in equine ASCs cells effected by metabolic syndrome via the regulation of *PERK*, *BiP*, *CHOP* and *eI2F-α* transcripts [23].

Previous reports indicated that persistent hyperglycemia was closely associated with a pro-inflammatory status, characterised by a significant elevation of IL-1β, IL-6, -1B and -8 and TNF-α in patients with uncontrolled diabetes and ketoacidosis [38]. Our experimental model of hyperglycemia resulted in a visible alteration of the HuASCs immunomodulatory properties and triggered an augmentation of the pro-inflammatory marker expression, including IL-1β and TNF-α at both the mRNA and the proteins levels. Additionally, the known anti-inflammatory abilities of HuASCs cells were markedly reduced as evidenced by the downregulation of main anti-inflammatory cytokine expression, namely *IL-10* and *TGF-β*. In the same way, earlier studies reported on the impaired immunomodulatory properties of ASCs cells in the course of metabolic disorders. *Aliakbari and colleagues* showed that ASCs isolated from patients affected by type 2 diabetes displayed an increased pro-inflammatory character, with an upregulated expression of pro-inflammatory factors and a concomitant declined expression of *IL-10* anti-inflammatory cytokine [39]. Additionally, *Rønningen* et al. postulated that elevated glucose epigenetically primed ASCs for upregulation of an inflammatory response [40]. Our preconditioning of HuASCs cells with calystegines sensibly promoted the proper anti-inflammatory ability of the cells, while modulating the engagement of the inflammatory response to a high glucose level via the regulation of pro- and anti-inflammatory cytokine expression at the mRNA and the protein levels. The anti-inflammatory effect of tested alkaloids has already been demonstrated in an in vivo model of mouse paw oedema induced by carrageenan [24]. Furthermore, the same alkaloids diminished hyperglycemia and hyperinsulinemia induced inflammation in HepG2 cells through the regulation of the NF- κB/JNK/TLR4 axis and the inhibition of downstream proinflammatory cytokines recruitment [29].

Little is known regarding pathways involved in inflammation, oxidative stress and ER stress in ASCs under hyperglycaemic conditions. Here, we recognized that calystegines protect HuASCs from the unfavorable hyperglycaemic microenvironment through a sustained AKT/PI3K/mTOR pathway. The phosphatidylinositol 3-kinase/protein kinase B/mammalian target of the rapamycin (PI3K/AKT/mTOR) signaling pathway has been shown to be involved in essential cellular functions, including cell proliferation, differentiation, metabolism, survival, cancer and insulin resistance [41]. Likewise, *Huang and colleagues*, presented evidence that identifying the therapeutic targets for AKT/PI3K/mTOR axis regulation may provide interesting options for obesity treatment [42]. Our outcomes showed that under the hyperglycaemic milieu, HuASCs were characterised by collapsed AKT/PI3K/mTOR signaling, as evidenced by the significant downregulation of *AKT*, *PI3K* and *mTOR* factors at the mRNA and the protein levels.

Hence, the AKT/PI3K/mTOR pathway has been reported to be strongly impaired in STZ-induced hyperglycemia and inflammation in Wistar rats [43]. Another recent study, pointed out that polymorphism in *PI3K*/*AKT*/*mTOR* genes were strongly associated with the risk of type 2 diabetes mellitus (T2DM) and its resulting hyperglycaemia [13]. In the present study, we were able to observe that calystegines applied as a preventive pre-treatment considerably enhanced the impaired AKT/PI3K/mTOR by preventing a decline in the expression of its main effectors at both the mRNA and the protein levels. As a matter of fact, the crosstalk between calystegines and AKT/Pi3K and mTOR has already been observed in our previous research, where the alkaloids efficiently ameliorated glucose metabolism and homeostasis in insulin resistance HepG2 cells mainly by the promotion of AKT/PI3K and SIRT/mTOR signalling cascades [29]. Thus, calystegines would improve the functions of ASCs cells through their ability to restore insulin signalling, confirming their utility as antidiabetic and antihyperglycaemic agents, which has even been proven in an in vivo model of STZ-induced diabetes in albino mice [44]. Thereby, modulation of AKT/PI3K/mTOR activity under hyperglycaemic conditions in ASCs may become a powerful therapeutic strategy for protecting stem cell pools against inflammation and oxidative and ER stress.

Notwithstanding the observed efficiency of calystegines in preventing hyperglycaemia-mediated HuASCs metabolic dysfunctions, the limitation of the presented research lies in the lack of in vivo experiment that would allow us to verify and to confirm that calystegines may substantially improve in situ ASCs properties.

## 5. Conclusions

The presented study outcomes showed that under hyperglycaemic conditions calystegines substantially protect human ASCs against apoptosis, inflammation and oxidative stress. Calystegines through the maintenance of the AKT/PI3K/mTOR pathway mitigate ER stress and therefore ensure ASCs survival and proper metabolic activity in a hyperglycaemic milieu, making calystegines a promising alternative for the effective management of metabolic conditions associated with hyperglycemia including diabetes mellitus, obesity and metabolic syndrome.

## Figures and Tables

**Figure 1 biomolecules-12-00460-f001:**
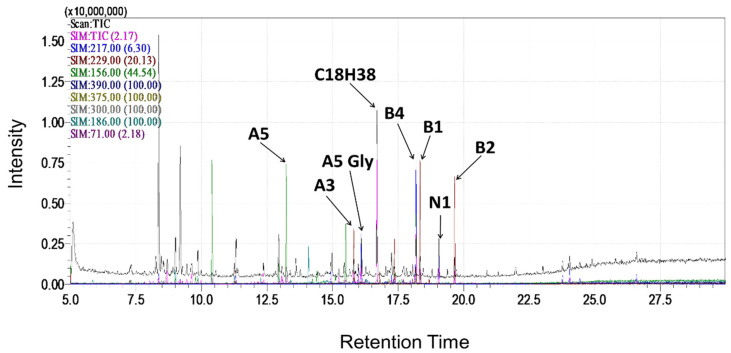
GC-MS chromatogram of *Hyoscyamus albus* seeds calystegines. Ions were monitorised in SIM. A total of 217 *m*/*z* was used as a common ion for calystegines B2, B3 and B4; 229 *m*/*z* for calystegine B2; 244 *m*/*z* for calystegine B4; 189 *m*/*z* for calystegine B1, 156 *m*/*z* for calystegines A3 and A5; 390 *m*/*z* for calystegine N1. Octadecan (C18H38, 71 *m*/*z*) was used as an internal standard [24].

**Figure 2 biomolecules-12-00460-f002:**
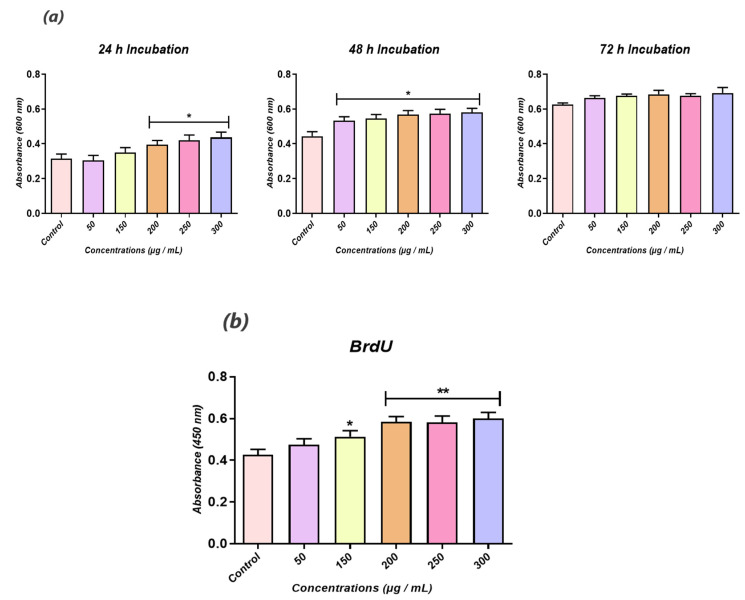
Biocompatibility assessment of total calystegines extract on HuASCs cells. (**a**) Histograms depicting the average absorbance at 600 nm of the metabolised Resazurin dye. (**b**) Average absorbances of incorporated BrdU into newly synthesised DNA of treated HuASCs cells. Representative data from three independent experiments are shown ± SD (n = 3). An asterisk (*) indicates a comparison of treated groups to untreated healthy cells. * *p* < 0.05, ** *p* < 0.01.

**Figure 3 biomolecules-12-00460-f003:**
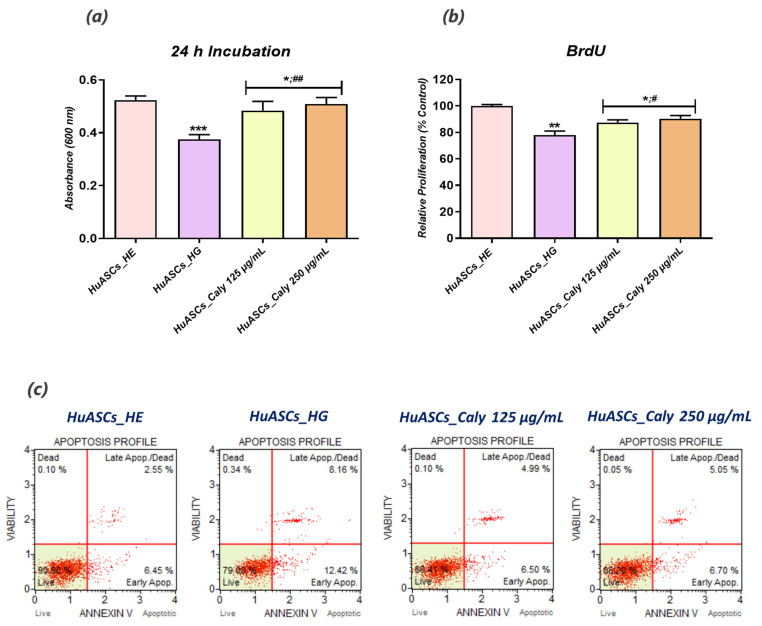
Pro-survival effect of total calystegine extract on HuASCs cells cultured under hyperglycemic conditions. (**a**) Histograms depicting the average absorbance at 600 nm of the metabolised Resazurin dye. (**b**) Average absorbances of incorporated BrdU into newly synthesised DNA of treated HuASCs cells. (**c**) Representative apoptosis dot plots from flow cytometry analysis. (**d**) Quantitative estimation of Annexin V/7-AAD positive and negative cells. (**e**) Relative gene expression quantitation of main apoptosis–associated marker levels. (**f**) Relative gene expression representation of different caspase transcripts. Representative data from three independent experiments are shown ± SD (n = 3). An asterisk (*) indicates a comparison of the HG group to untreated healthy cells. A hashtag (#) indicates a comparison of the HG group pre-treated with calystegines to the HG untreated healthy cells. */# *p* < 0.05, **/## *p* < 0.01, ***/### *p* < 0.001. HuASCs_HE: human ASCs healthy untreated cells; HuASCs_HG: Hyperglycaemic human ASCs cells exposed to a high concentration of glucose. HuASCs_Caly 125 µg/mL: Hyperglycaemic human ASCs cells pre-treated with 125 μg/mL calystegines extracted and exposed to a high concentration of glucose. HuASCs_Caly 250 µg/mL: Hyperglycaemic human ASCs cells pre-treated with 250 μg/mL calystegines extracted and exposed to a high concentration of glucose.

**Figure 4 biomolecules-12-00460-f004:**
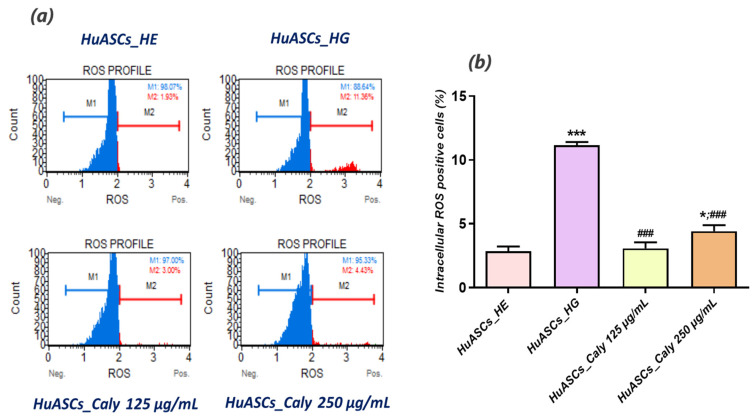
Antioxidant effect of total calystegine extract on HuASCs cells cultured under hyperglycaemic conditions. (**a**) Representative dot plots for ROS^-^/ROS^+^ cells distribution. (**b**) Histograms depicting average ROS positive cells. (**c**) Relative gene expression average of endogenous antioxidant enzymes transcripts. (**d**) SOD and CAT enzymatic activities. Representative data from three independent experiments are shown ± SD (n = 3). An asterisk (*) indicates a comparison of the HG group to untreated healthy cells. A hashtag (#) indicates a comparison of the HG group pre-treated with calystegines to the HG untreated healthy cells. */# *p* < 0.05, **/## *p* < 0.01, ***/### *p* < 0.001. HuASCs_HE: human ASCs healthy untreated cells; HuASCs_HG: hyperglycaemic human ASCs cells exposed to a high concentration of glucose. HuASCs_Caly 125 µg/mL: hyperglycaemic human ASCs cells pre-treated with 125 μg/mL calystegines extracted and exposed to a high concentration of glucose. HuASCs_Caly 250 µg/mL: hyperglycaemic human ASCs cells pre-treated with 250 μg/mL calystegines extracted and exposed to a high concentration of glucose.

**Figure 5 biomolecules-12-00460-f005:**
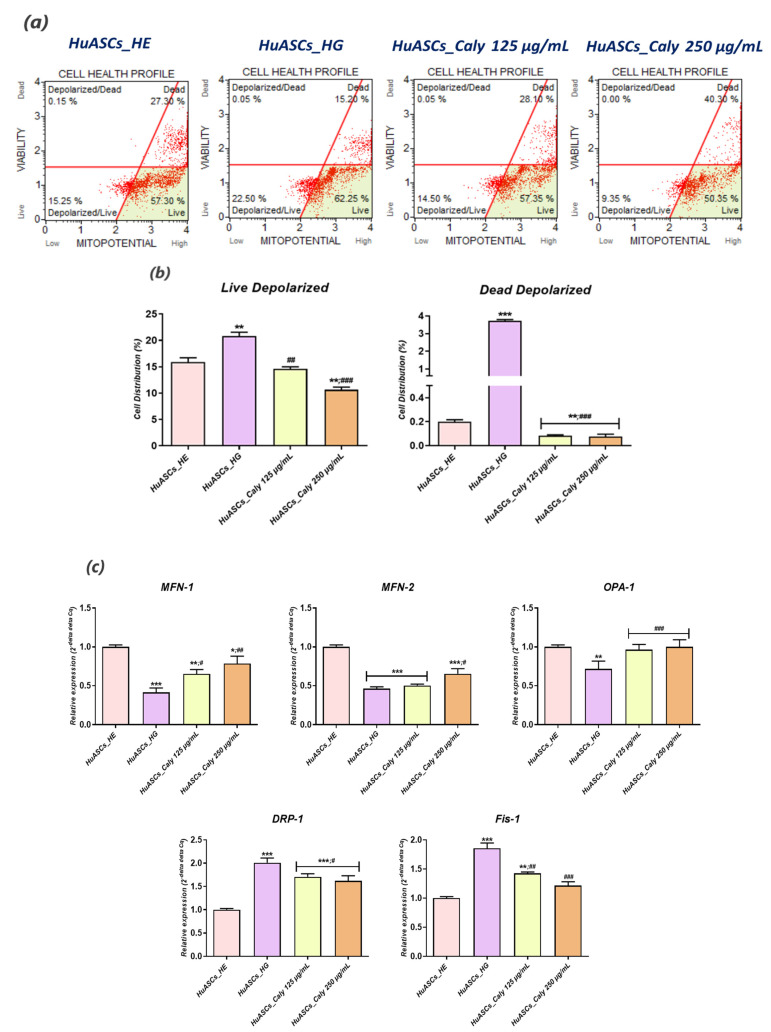
Beneficial effects of total calystegine extract on abnormal mitochondrial dynamics in HuASCs cells cultured under hyperglycaemic conditions. (**a**) Representative dot plots for MUSE MitoPotential analysis. (**b**) Percentage of live and dead cells exhibiting depolarized mitochondria. (**c**) Relative gene expression of mitochondrial fusion and fission associated genes. (**d**) Epi-fluorescent confocal microscope micrographs of MitoRed stained cells; scale bar size 18 μm. Representative data from three independent experiments are shown ± SD (n = 3). An asterisk (*) indicates a comparison of the HG group to the untreated healthy cells. A hashtag (#) indicates a comparison of HG group pre-treated with calystegines to the HG untreated healthy cells. */# *p* < 0.05, **/## *p* < 0.01, ***/### *p* < 0.001. HuASCs_HE: human ASCs healthy untreated cells; HuASCs_HG: hyperglycaemic human ASCs cells exposed to a high concentration of glucose. HuASCs_Caly 125 µg/mL: hyperglycaemic human ASCs cells pre-treated with 125 μg/mL calystegines extracted and exposed to a high concentration of glucose. HuASCs_Caly 250 µg/mL: hyperglycaemic human ASCs cells pre-treated with 250 μg/mL calystegines extracted and exposed to a high concentration of glucose.

**Figure 6 biomolecules-12-00460-f006:**
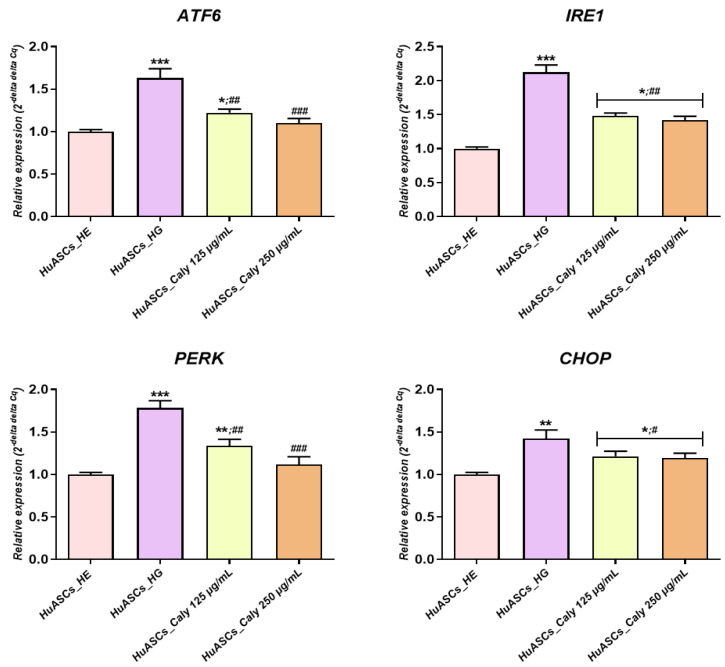
Preventive effect of total calystegines extract on ER stress engagement in HuASCs cells cultured in a hyperglycaemic milieu. Relative gene expression of ER stress master sensors and effectors. Representative data from three independent experiments are shown ± SD (n = 3). An asterisk (*) indicates a comparison of the HG group to the untreated healthy cells. A hashtag (#) indicates a comparison of the HG group pre-treated with calystegines to the HG untreated healthy cells. */# *p* < 0.05, **/## *p* < 0.01, ***/### *p* < 0.001. HuASCs_HE: human ASCs healthy untreated cells; HuASCs_HG: hyperglycaemic human ASCs cells exposed to a high concentration of glucose. HuASCs_Caly 125 µg/mL: hyperglycaemic human ASCs cells pre-treated with 125 μg/mL calystegines extracted and exposed to a high concentration of glucose. HuASCs_Caly 250 µg/mL: hyperglycaemic human ASCs cells pre-treated with 250 μg/mL calystegines extracted and exposed to a high concentration of glucose.

**Figure 7 biomolecules-12-00460-f007:**
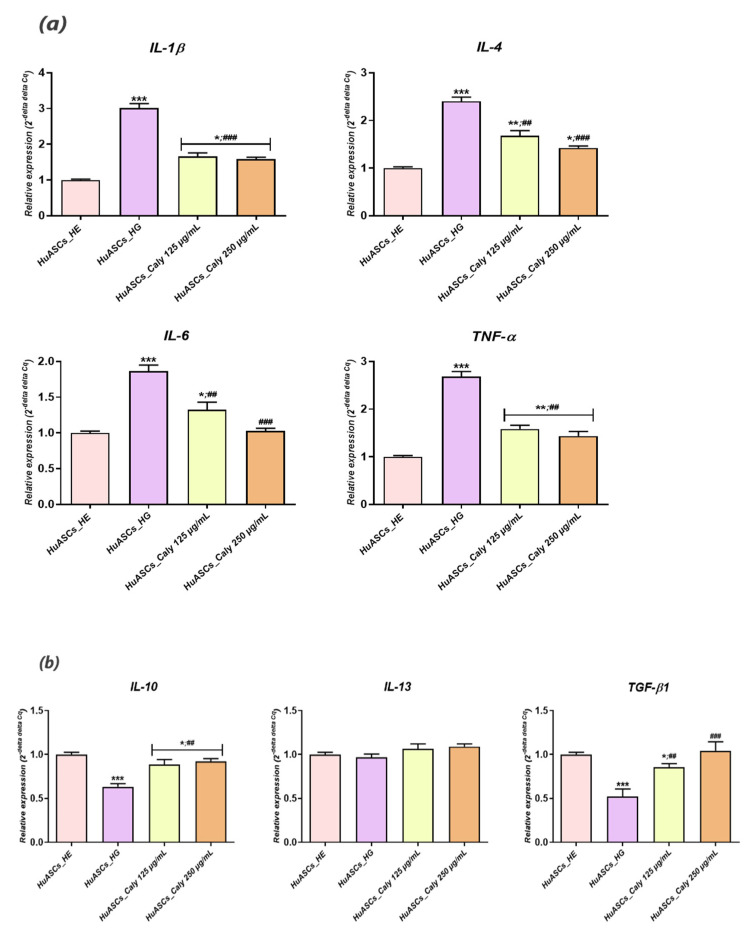
Reducing effect of total calystegine extract on hyperglycemia-mediated inflammatory response in HuASCs cells. (**a**) Histograms summarising the relative gene expression of key pro-inflammatory cytokines. (**b**) Representation of normalized gene expression of main anti-inflammatory markers. (**c**) ELISA quantification results of IL-1β and TNF-α proteins levels. Representative data from three independent experiments are shown ± SD (n  =  3). An asterisk (*) indicates a comparison of the HG group to the untreated healthy cells. A hashtag (#) indicates a comparison of HG group pre-treated with calystegines to HG untreated healthy cells. */# *p* < 0.05, **/## *p* < 0.01, ***/### *p* < 0.001. HuASCs_HE: human ASCs healthy untreated cells; HuASCs_HG: hyperglycaemic human ASCs cells exposed to a high concentration of glucose. HuASCs_Caly 125 µg/mL: hyperglycaemic human ASCs cells pre-treated with 125 μg/mL calystegines extracted and exposed to a high concentration of glucose. HuASCs_Caly 250 µg/mL: hyperglycaemic human ASCs cells pre-treated with 250 μg/mL calystegines extracted and exposed to a high concentration of glucose.

**Figure 8 biomolecules-12-00460-f008:**
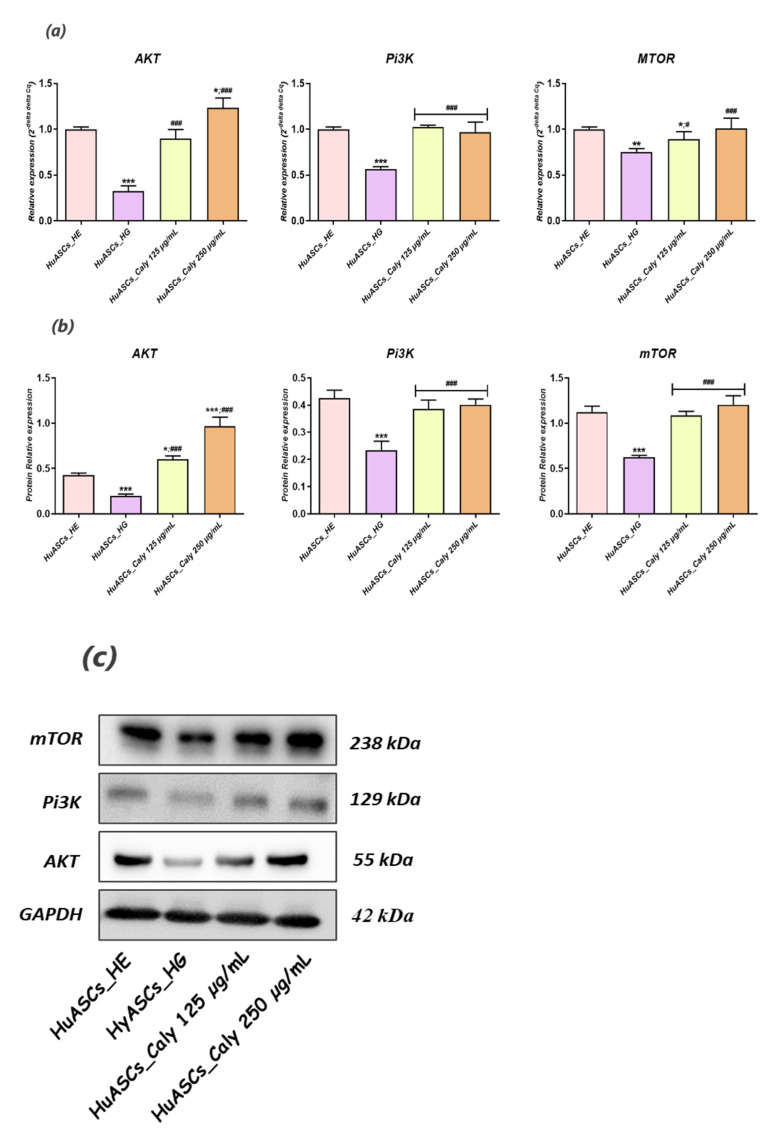
Promoting effect of total calystegine extract on hyperglycemia-mediated Akt/Pi3K/mTOR signaling pathway failure in HuASCs cells. (**a**) Histograms summarizing the relative gene expression of AKT, PI3K and MTOR mRNAs. (**b**) Relative Akt, Pi3K and mTOR protein expression levels normalized to GAPDH housekeeping protein and evaluated using western blot. (**c**) Representative immunoblots for each assayed protein detected by chemiluminescence. Representative data from three independent experiments are shown  ±  SD (n  =  3). An asterisk (*) indicates a comparison of the HG group to the untreated healthy cells. A hashtag (#) indicates a comparison of the HG group pre-treated with calystegines to the HG untreated healthy cells. */# *p* < 0.05, ** *p* < 0.01, ***/### *p* < 0.001. HuASCs_HE: Human ASCs healthy untreated cells; HuASCs_HG: hyperglycaemic human ASCs cells exposed to a high concentration of glucose. HuASCs_Caly 125 µg/mL: hyperglycaemic human ASCs cells pre-treated with 125 μg/mL calystegines extracted and exposed to a high concentration of glucose. HuASCs_Caly 250 µg/mL: hyperglycaemic human ASCs cells pre-treated with 250 μg/mL calystegines extracted and exposed to a high concentration of glucose.

**Table 1 biomolecules-12-00460-t001:** Primers used for gene expression analysis.

Gene	Primer	Sequence 5′–3′	Amplicon Length (bp)	Accession No.
*p53*	F: R:	AGATAGCGATGGTCTGGC TTGGGCAGTGCTCGCTTAGT	381	NM_001126118.1
*p21*	F: R:	AGAAGAGGCTGGTGGCTATTT CCCGCCATTAGCGCATCAC	169	NM_001220777.1
*BAX*	F: R:	ACCAAGAAGCTGAGCGAGTGTC ACAAAGATGGTCACGGTCTGCC	356	XM_011527191.1
*Bcl2*	F: R:	ATCGCCCTGTGGATGACTGAG CAGCCAGGAGAAATCAAACAGAGG	129	NM_000633.2
*Casp3*	F: R:	CTCTGGTTTTCGGTGGGTGT CTTCCATGTATGATCTTTGGTTCC	136	NM_004346.4
*Casp9*	F: R:	CAGGCCCCATATGATCGAGG CTGGCCTGTGTCCTCTAAGC	142	NM_032996.3
*Casp7*	F: R:	AGGGGACTGTTTTCAGATGGAG AACGCCCATACCTGTCACTT	177	NM_001267058.2
*Casp8*	F: R:	TGCTGAGCACGTGGAGTTAG CAGGCTCAGGAACTTGAGGG	178	NM_001080125.2
*Casp6*	F: R:	TCATGAGAGGTTCTTTTGGCAC CACACACAAAGCAATCGGCA	197	NM_001226.4
*Sod1*(*Cu*/*Zn SOD*)	F: R:	CATTCCATCATTGGCCGCAC GAGCGATCCCAATCACACCA	130	NW_001867397.1
*Sod2* (*Mn SOD*)*:*	F: R:	GGACAAACCTGAGCCCCAAT TTGGACACCAGCCGATACAG	125	NW_001867408.1
*Cat*	F: R:	ACCAAGGTTTGGCCTCACAA TTGGGTCAAAGGCCAACTGT	112	XM_014851065.1
*GPx*	F: R:	TCCGGGACTACACCCAGATG TCTTGGCGTTCTCCTGATGC	108	NM_000581.4
*Mfn1*	F: R:	GTTGCCGGGTGATAGTTGGA TGCCACCTTCATGTGTCTCC	146	NM_033540.3
*Mfn2*	F: R:	AATCTGAGGCGACTGGTGAC GCTGCACTGTACCCTGAGTT	294	NM_013987.3
*OPA-1*	F: R:	TGCCTGACATTGTGTGGGAAA TTCCGGAGAACCTGAGGTAA	161	NM_015560.3
*DRP-1*	F: R:	GGTGAACCCGTGGATGATAAA CCTCAGGCACAAATAAAGCAG	265	NM_001278465.2
*Fis-1*	F: R:	TGGTGCGGAGCAAGTACAAT TGCCCACGAGTCCATCTTTC	252	NM_016068.3
*ATF-6*	F: R:	ACCTCCTTGTCAGCCCCTAA CACTCCCTGAGTTCCTGCTG	230	NM_007348.4
*IRE-1*	F: R:	CGGCCTCGGGATTTTTGGA AGAAAGGCAGGCTCTTCCAC	177	NM_001433.5
*PERK*	F: R:	TGCTCCCACCTCAGCGAC TTTCAGGATCCAAGGCAGCA	212	NM_004836.7
*CHOP*	F: R:	GTTAAAGATGAGCGGGTGGC TGCAGTTGGATCAGTCTGCTT	124	NM_001195057.1
*IL-1β*	F: R:	AAACAGATGAAGTGCTCCTTCCAG TGGAGAACACCACTTGTTGCTCCA	391	NM_000576.3
*IL-4*	F: R:	CTTTGCTGCCTCCAAGAACAC GCGAGTGTCCTTCTCATGGT	94	NM_000589.4
*IL-6*	F: R:	GCGAGTGTCCTTCTCATGGT TCCTTCTCCACAAACATGTAACAA	319	NM_001318095.2
*TNF-* *α*	F: R:	AGTGACAAGCCTGTAGCCCA GTCTGGTAGGAGACGGCGAT	242	NM_000594.4
*IL-10*	F: R:	AGACAGACTTGCAAAAGAAGGC TCGAAGCATGTTAGGCAGGTT	148	NM_000572.3
*IL-13*	F: R:	GCAATGGCAGCATGGTATGG AAGGAATTTTACCCCTCCCTAACC	385	NM_001354993.2
*TGF-β1*	F: R:	ACTCGCCAGAGTGGTTATCT GGTAGTGAACCCGTTGATGT	152	NM_000660.7
*Akt*	F: R:	CTGTCATCGAACGCACCT GTCTGGATGGCGGTTGTC	178	NM_005163.2
*Pi3k*	F: R:	TTTAATCTGCCAGGCGGAGG CCAGAATTCCATGGGGCAGT	151	NM_006218.4
*MTOR*	F: R:	GAACCTCAGGGCAAGATGCT CTGGTTTCCTCATTCCGGCT	125	NM_004958.4
*GAPDH*	F: R:	GTCAGTGGTGGACCTGACCT CACCACCCTGTTGCTGTAGC	256	NM_001289746.1

P53: tumor suppressor p53; p21: Cyclin-dependent kinase inhibitor 1; BAX: BCl-2 associated X protein; Bcl-2: B-cell lymphoma 2; Casp3: Caspase 3; Casp9: Caspase 9; Casp7: Caspase 7; Casp8: Caspase 8; Casp6: Caspase 6; Sod1: Sod1 (Cu/Zn SOD): Copper-zinc-dependant superoxide dismutase (CuZnSOD); Sod2 (Mn SOD): Manganese-dependent superoxide dismutase (MnSOD); CAT: Catalase; GPx: Glutathione Peroxidase; Mfn1: Mitofusin 1; Mfn: Mitofusin 2; OPA-1: OPA1 Mitochondrial Dynamin Like GTPase; DRP-1: Interaction with the effector dynamin-related protein 1; Fis-1: Mitochondrial fission 1 protein; ATF-6: activating transcription factor 6; IRE-1: endoplasmic reticulum to nucleus signaling 1; PERK: eukaryotic translation initiation factor 2 alpha kinase 3 (EIF2AK3); CHOP: DNA damage inducible transcript 3 (DDIT3); IL-1β: Interleukin 1 beta; IL-4: Interleukin 4; IL-6: Interleukin 6; TNF-α: Tumor Necrosis Factor alpha; IL-10: Interleukine 10; IL-13: Interleukine 13; TGF-β1: transforming growth factor beta 1; Akt1: Serine/threonine 308 Kinase 1; Pi3k: Phosphoinositide 3-Kinase; mTOR: mechanistic target of rapamycin.

**Table 2 biomolecules-12-00460-t002:** List of antibodies employed for protein profiling using western blot analysis.

Antibody	Dilution	Catalogue No.
Akt Pan	1:1000	Invitrogen, 44-609G
Pi3kCD	1:1000	Invitrogen, PA5-83748
mTOR	1:500	Novus Biologicals, nb100-240
GAPDH	1:2000	Biorbyt, orb323277

Akt: Protein Kinase B; Pi3kCD: Phosphatidylinositol 3-Kinase; wTOR, Target of rapamycin/Mammalian Target of rapamycin.

## Data Availability

The data that support the findings of this study are available from the corresponding author, upon reasonable request.

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
