# Peer review of "Calystegines Improve the Metabolic Activity of Human Adipose Derived Stromal Stem Cells (ASCs) under Hyperglycaemic Condition through the Reduction of Oxidative/ER Stress, Inflammation, and the Promotion of the AKT/PI3K/mTOR Pathway"

_biomolecules, 2022, doi:10.3390/biom12030460_

Round 1
Reviewer 1 Report
In the manuscript by Anna Kowalczuk et al, the authors demonstrated the protection of calystegines in the Human ASCs cells cultured with high glucose. It indicates calystegines may be helpful in the treatment of glucotoxicity induced by hyperglycaemia. In detail, calystegines is able to decrease the ASCs cell apoptosis induced by high glucose (30mM) and the protection may be from the decreased oxidative and ER stress and inflammation. The authors detected the transcription level of the genes associated with these processes by qPCR and showed the relief of oxidative and ER stress and inflammation with the treatment of calystegines. The underlying mechanism may be related to the change of PI3K/AKT/mTOR metabolic-associated axis. In the manuscript, it is clear that the improved ASCs survival with the treatment of calystegines. However, the underlying mechanism related to the phenotype needs to be further validated. Here are my major concerns:
- Although the transcription change in ASCs treated with calystegines is proved by qPCR, the protein related to ER stress should be detected by WB including BIP, CHOP, ATF4, or ATF6. The cleaved ATF6 serves as a better marker for the induction of ER stress than mRNA. For the Pi3K/AKT/mTOR metabolic-associated axis, the mRNA level of these genes or the total level of the proteins is not enough to answer if the signaling pathway is activated or not. The phosphorylation level of the proteins is more important for the activation of the signaling. For example, the phosphorylation site at Thr308 or Ser473 for AKT should be detected. For mitochondrial fusion, more evidences may be provided such as mitochondria staining if possible.
- It seems the survival protection of calystegines is related to the decreased oxidative and ER stress and inflammation. However, the logic between these processes is hard to tell to rescue the cell survival treated with high glucose. Here it is recommended to use some specific chemicals to rescue the process induced by high glucose. 4-PBA could be used as a control to know if ER stress is critical for cell survival rescue. NAC could be used to inhibit ROS induction to see how the mitochondria function is related to cell survival treated with high glucose.
- It would be better if there are in vivo data for the treatment of diabetes with calystegines.
Here are the minor concerns:
- The quality of calystegines used in the experiment should be presented such as GC-MS map.
- Line 126-129 are repeated with Line 131-134.
- Line 162: 0.5%
- Line 160, 165: cm2
- Line 265, 278: 1*106
- Line335: mTOR
- Line 283, 459: redundant space
- Line 603: redundant -
Author Response
Dear Editor and Reviewer #1,
Thank you very much for all of your pertinent comments and the time you spent to review our work. Please find below the responses to your remarks
Reviewer#1
- Comment: Although the transcription change in ASCs treated with calystegines is proved by qPCR, the protein related to ER stress should be detected by WB including BIP, CHOP, ATF4, or ATF6. The cleaved ATF6 serves as a better marker for the induction of ER stress than mRNA.
- Response: We totally agree with the reviewer suggestion; performing a protein profiling always brings more value and more information about pathways regulation. Unfortunately, we are currently unable to perform any additional western blot analysis due to financial limitations. Moreover, it has been previously assumed that mRNA concentrations of ER chaperons, ERAD components, such as BiP, HERP, WARS, PDIA4, P58IPK, EDEM1, XBP1, and ATF4, or genes, that are involved in the induction of apoptosis, such as Chop or caspases, are reliable markers of ER stress. Moreover, other study demonstrated that Knockdown of CHOP attenuated MMC-induced apoptosis by increasing the ratio of BCL-2/BAX and decreasing BIM expression, suggesting that ER stress is involved in MMC-induced fibroblast apoptosis. Interestingly, knockdown of PERK significantly decreased ER stress-mediated apoptosis by reducing the expression of CHOP, BIM and cleaved caspase-3., resulting in an enhancement in cell viability, which implies reliable correlation between ER stress markers expression at mRNA level and UPR.
- Samali, A., FitzGerald, U., Deegan, S., & Gupta, S. (2010). Methods for Monitoring Endoplasmic Reticulum Stress and the Unfolded Protein Response. International Journal of Cell Biology, 2010, 1–11. doi:10.1155/2010/830307.
- Shi, K., Wang, D., Cao, X., & Ge, Y. (2013). Endoplasmic Reticulum Stress Signaling Is Involved in Mitomycin C(MMC)-Induced Apoptosis in Human Fibroblasts via PERK Pathway. PLoS ONE, 8(3), e59330. doi:10.1371/journal.pone.0059330.
- Gessner, D. K., Schlegel, G., Ringseis, R., Schwarz, F. J., & Eder, K. (2014). Up-regulation of endoplasmic reticulum stress induced genes of the unfolded protein response in the liver of periparturient dairy cows. BMC Veterinary Research, 10(1), 46. doi:10.1186/1746-6148-10-46.
- Comment: For the Pi3K/AKT/mTOR metabolic-associated axis, the mRNA level of these genes or the total level of the proteins is not enough to answer if the signaling pathway is activated or not. The phosphorylation level of the proteins is more important for the activation of the signaling. For example, the phosphorylation site at Thr308 or Ser473 for AKT should be detected. For mitochondrial fusion, more evidences may be provided such as mitochondria staining if possible.
- Response: thank you for this suggestion; we totally agree with the reviewer that showing the phosphorylation patterns of AKT, Pi3K and mTOR would bring more precise insights in the influence of calystegines on Pi3K/AKT/mTOR signalling pathway under hyperglycaemic milieu. However, as mentioned in the first comment, we are currently unable to carry additional proteins analysis due to financial issues. Thus, in the present study, we showed that calystegines ameliorate expression of the aforementioned mediators at both mRNA and protein levels, which represents an interesting preliminary finding that may lead to consider other more in-depth studies. Furthermore, in a recently published study, Swiderska et al., similarly demonstrated that chronic and Intermittent hyperglycemia modulates Expression of PI3K/AKT Pathway in human visceral adipocytes at both mRNA and protein levels. However, taking into account the reviewer comment, we included a “Study Limitations” section to our manuscript. We also included MitoRed stained cells photomicrographs for mitochondrial network visualization.
- Åšwiderska, E., Strycharz, J., Wróblewski, A., Czarny, P., Szemraj, J., Drzewoski, J., & ÅšliwiÅ„ska, A. (2021). Chronic and Intermittent Hyperglycemia Modulates Expression of Key Molecules of PI3K/AKT Pathway in Differentiating Human Visceral Adipocytes. International Journal of Molecular Sciences, 22(14), 7712. Doi:10.3390/ijms22147712.
- Comment: It seems the survival protection of calystegines is related to the decreased oxidative and ER stress and inflammation. However, the logic between these processes is hard to tell to rescue the cell survival treated with high glucose. Here it is recommended to use some specific chemicals to rescue the process induced by high glucose. 4-PBA could be used as a control to know if ER stress is critical for cell survival rescue. NAC could be used to inhibit ROS induction to see how the mitochondria function is related to cell survival treated with high glucose.
- Response: that you for this interesting suggestion. The crosstalk between Hyperglycaemia, Oxidative stress, Mitochondrial dysfunction and apoptosis has been extensively studied and demonstrated. Several signaling pathways can be altered by having hyperglycemia in different tissues, producing oxidative stress, the formation of advanced glycation end products (AGEs), as well as the secretion of the pro-inflammatory cytokines and cellular death (pathological autophagy and/or apoptosis). However, the signaling pathways that are directly triggered by hyperglycemia appear to have a pivotal role in diabetic complications due to the production of reactive oxygen species (ROS), oxidative stress, and cellular death.
Chronic hyperglycemia is characteristic of diabetes and the main attributor to the multiple complications associated with the disease. It is well established that oxidative stress plays a key role in the onset and development of this condition. Glucose is the main fuel used by mitochondria to obtain energy by electron transport chain (ETC), a process of which ROS production is a by-product. High glucose concentrations can saturate antioxidant defences and induce oxidative stress in mitochondria via the increased ROS generation. High ROS can overburden and saturate antioxidant mechanisms, thus leading to oxidative stress in the ER as well. mtDNA in turn, is highly vulnerable to ROS-induced damage, which can result in mutations and the production of altered ETC proteins. This results in the malfunctioning of ETC and further generation of ROS, thereby accentuating oxidative stress in mitochondria. In turn, oxidative stress in the ER leads to the accumulation of unfolded proteins in the ER lumen, which, in turn, increases oxidative stress. However, perpetuated state of oxidative stress, unfolded proteins accumulation and mitochondrial and ER dysfunction progressively impairs cellular functions and ultimately leads to the activation of apoptotic pathways, thus increasing their death rate. Impaired mitochondria release cytochrome c in the cytosol that activates apoptotic pathways; similarly, ER stress activates the C/EBP homologous protein (CHOP) protein which is a master apoptosis initiator. CHOP further downregulate the anti-apoptotic protein Bcl-2 and alter the redox state of the cell.
To this end, different therapeutic molecules have been found to mediate anti-apoptotic action through multiple molecular mechanisms including the inhibition of oxidative stress, ER stress and inflammation. Glycemic controls, in conjunction with modulation of PKC and/or NADPH-oxidase for example, downregulate the pro-inflammatory cytokines, leading to a reduced amount of ROS, and, consequently, decreased cellular death.
So, we think that it is reasonable to postulate that pro-survival effect of calystegines maybe related to its antioxidative effect as well as its ability to alleviate ER stress, mitochondrial dysfunction and inflammation.
- Comment: It would be better if there are in vivo data for the treatment of diabetes with calystegines.
- Response: Thank you for the suggestion. In vivo experiment would obviously confirm if the observed in vitro effects maybe positively translated to in vivo Unfortunately, we do not process either any ethical approval to conduct such experiments, neither the necessary facilities for animal experimentation. We would like however to emphasize that effect of calystegines on an in-vivo model of streptozotocin-induced diabetes as well as inflammation on albino mice has already been conduction previously by our group and data are already published.
- Bourebaba L, Saci S, Touguit D, Gali L, Terkmane S, Oukil N, Bedjou F. Evaluation of antidiabetic effect of total calystegines extracted from Hyoscyamus albus. Biomed Pharmacother. 2016 Aug;82:337-44. doi: 10.1016/j.biopha.2016.05.011.
- Bourebaba, L., Sullini, G., Mendiola, J. A., Bourebaba, Y., Deghima, A., Oukil, N., & Bedjou, F. (2016). In-vivo edema inhibition of Hyoscyamus albus antioxidant extracts rich in calystegines. Industrial Crops and Products, 89, 316–322. doi:10.1016/j.indcrop.2016.04.067.
- Comment: The quality of calystegines used in the experiment should be presented such as GC-MS map.
- Response: GC-MS chromatogram has been added.
- Comment: Line 126-129 are repeated with Line 131-134; Line 162: 0.5%; Line 160, 165: cm2; Line 265, 278: 1*106; mTOR; Line 283, 459: redundant space; Line 603: redundant.
- Response: All the highlighted errors have been corrected.
Reviewer 2 Report
This is an interesting paper that contains novel findings that could lead to new therapeutic approaches for hyperglycemic metabolic conditions. The manuscript is well-structured and written in general. References were also properly cited.
However, there are a number of issues that the authors must address before the manuscript can be accepted.
-Abstract Line 13: “Diabetes mellitus (DM)” instead of “DM”
-Abstract Line 20: “…and sensibly diminished” Revise. May be the authors mean significantly?
-Abstract Line 26: “possess” instead of “process”
-M&M section requires careful revision because many paragraphs are duplicated: Lines 131-134; Lines 262-273;
-Line 134: use italics to refer to “Hyoscyamus albus”
-Lines 145-147: The chemical characterization of the obtained plant extract must be improved (percentage)
-Line 162 “0.5%” instead of “0,5%”
- Please revise Section 2.4 to improve clarity. Proliferation is not the same as viability. In addition, some sentences in section 2.7 are duplicated.
-Line 198 “24h” instead of “24”
-Line 209 “diluted BrdU reagent was added” instead of “diluted the BrdU reagent was added”
-Remove lines 262-273.
-On line 364, remove *** p 0.001 because it is not applicable here.
- Line 379: “two tested concentrations” instead of “two tested doses”
-Line 383 “…as well as caspases transcripts (Figure 2.e and 3.f)” There is no Figure 3.f in the manuscript.
-Line 384 “…visible restoration of the Bcl-2 cell survival gene to reach a basal level” Calystegines were added prior to the stimulus (HG), so they avoided a change in the Bcl-2 cell survival gene rather than restoring to a basal level.
- The results shown in Figure 2.d - late apoptotic cells - do not correspond to the representative apoptosis dot plots from flow cytometry analysis, which show a higher value for HuASCs Caly125.
-Line 395 “(f) Relative gene expression representation of different caspases transcripts.” This figure is not included in the manuscript.
-Line 417 revise “Sod1, Sod1 and Cat transcripts”
-Line 420 “…correlates with the observed remarkable drop in the intrinsic activities of the two SOD 420 and CAT enzymes in the HG-nontreated HuASCs cells (Figure 3.d).” Again, these findings are not included in the manuscript. The same can be said for lines 428 and 434.
-Line 460 “in too excessive mitochondrial ROS”
-Line 575: correct to “(c)”
Discussion:
- Calystegines were used prior to inducing hyperglycemia in the current study. The authors should discuss whether these results could be seen if applied after, as this would be more clinically relevant.
-Revise sentence line 601-605
-Line 624 “whether” instead of “weather”
Line 627 “…and restored endogenous SOD and CAT antioxidant enzymatic activities” Calystegines were added prior to the stimulus (HG), so they avoided a change in the antioxidant enzymes rather than restoring to a basal level. The same can be said for line 701.
-Line 633 “calystegines possess” instead of “calystegines processes”
-Line 672 “affected” instead of “effected”
Line 729 “calystegines” instead of “calyatsegines”
-The conclusion section could use some English polishing.
The manuscript would benefit greatly from an English revision to correct typos and punctuation errors.
Author Response
Dear Editor and Reviewer #2,
Thank you very much for all of your pertinent comments and the time you spent to review our work. Please find below the responses to your remarks
Reviewer#2
- Comment: Abstract Line 13: “Diabetes mellitus (DM)” instead of “DM”
- Response: “DM” has been replaced with “Diabetes mellitus (DM)” as required.
- Comment: Abstract Line 20: “…and sensibly diminished” Revise. May be the authors mean significantly?
- Response: the sentence has been revised; “sensibly” has been replaced with “significantly”.
- Comment: Abstract Line 26: “possess” instead of “process”
- Response: the sentence has been corrected.
- Comment: M&M section requires careful revision because many paragraphs are duplicated: Lines 131-134; Lines 262-273.
- Response: M&M section has been revised and edited.
- Comment: Line 134: use italics to refer to “Hyoscyamus albus”.
- Response: plant name has been edited.
- Comment: Lines 145-147: The chemical characterization of the obtained plant extract must be improved (percentage).
- Response: The phytochemical composition representation of albus has been improved, GC-MS chromatogram has been included.
- Comment: Line 162 “0.5%” instead of “0,5%”
- Response: Corrected.
- Comment: Please revise Section 2.4 to improve clarity. Proliferation is not the same as viability. In addition, some sentences in section 2.7 are duplicated.
- Response: 2.4 section has been carefully revised and improved.
- Comment: Line 198 “24h” instead of “24”
- Response: Corrected.
- Comment: Line 209 “diluted BrdU reagent was added” instead of “diluted the BrdU reagent was added”
- Response: Corrected
- Comment: Remove lines 262-273.
- Response: the designated lines have been removed.
- Comment: On line 364, remove *** p 0.001 because it is not applicable here.
- Response: the *** p 0.001 has been deleted.
- Comment: Line 379: “two tested concentrations” instead of “two tested doses”
- Response: Corrected.
- Comment: Line 383 “…as well as caspases transcripts (Figure 2.e and 3.f)” There is no Figure 3.f in the manuscript.
- Response: The Figure 3.f has been included.
- Comment: Line 384 “…visible restoration of the Bcl-2 cell survival gene to reach a basal level” Calystegines were added prior to the stimulus (HG), so they avoided a change in the Bcl-2 cell survival gene rather than restoring to a basal level.
- Response: Thank you for this observation. The sentence has been corrected, as we indeed evaluated the protective effect of calystegines.
- Comment: The results shown in Figure 2.d - late apoptotic cells - do not correspond to the representative apoptosis dot plots from flow cytometry analysis, which show a higher value for HuASCs Caly125.
- Response: The dot plots shown in Figure 2.d are only one representative of the analysis which has been repeated three times. To avoid any confusion, the dot plot has been replaced with a more suitable representative.
- Comment: Line 395 “(f) Relative gene expression representation of different caspases transcripts.” This figure is not included in the manuscript.
- Response: Thank you very much for this comment. We noticed that some of the presented figures disappeared during manuscript editing process. We now included all the necessary results and figures.
- Comment: Line 417 revise “Sod1, Sod1 and Cat transcripts”
- Response: Revised
- Comment: Line 420 “…correlates with the observed remarkable drop in the intrinsic activities of the two SOD 420 and CAT enzymes in the HG-nontreated HuASCs cells (Figure 3.d).” Again, these findings are not included in the manuscript. The same can be said for lines 428 and 434.
- Response: We included all the results now. Some figures were lost during the manuscript editing to the journal’s template form.
- Comment: Line 460 “in too excessive mitochondrial ROS”
- Response: Corrected.
- Comment: Line 575: correct to “(c)”
- Response: Corrected.
- Comment: Calystegines were used prior to inducing hyperglycemia in the current study. The authors should discuss whether these results could be seen if applied after, as this would be more clinically relevant.
- Response: thank you very much for this interesting suggestion. While the current study was dedicated in the evaluation of the protective effect of calystegines on hyperglycaemia induced human ASCs dysfunction, previous study also demonstrated that calystegines administrated to diabetic mice (after diabetes induction) for a period of 20 days similarly reduced the blood glucose level and resulted in a regeneration of islets of Langerhans, we can assure that calystegines may exert similar effects if they were applied after HG induction. However, this is a very interesting concept that we can consider in future experiments.
- Bourebaba L, Saci S, Touguit D, Gali L, Terkmane S, Oukil N, Bedjou F. Evaluation of antidiabetic effect of total calystegines extracted from Hyoscyamus albus. Biomed Pharmacother. 2016 Aug;82:337-44. doi: 10.1016/j.biopha.2016.05.011.
- Comment: Revise sentence line 601-605.
- Response: Sentence revised.
- Comment: Line 624 “whether” instead of “weather”.
- Response: Corrected.
- Comment: Line 627 “…and restored endogenous SOD and CAT antioxidant enzymatic activities” Calystegines were added prior to the stimulus (HG), so they avoided a change in the antioxidant enzymes rather than restoring to a basal level. The same can be said for line 701.
- Response: The referred sentences have been corrected.
- Comment: Line 633 “calystegines possess” instead of “calystegines processes”
- Response: Corrected.
- Comment: Line 672 “affected” instead of “effected”
- Response: Corrected
- Response: Line 729 “calystegines” instead of “calyatsegines”
- Response: Corrected
- Comment: The conclusion section could use some English polishing.
- Response: The conclusion has been improved.
- Comment: The manuscript would benefit greatly from an English revision to correct typos and punctuation errors.
Response: The manuscript has been edited and improved
Round 2
Reviewer 1 Report
Although no more experiment is done following my suggestion, the author's responses claimed their difficulties due to financial issues and has addressed part of my corcerns about the experiments design or methods used in the manuscript.
Reviewer 2 Report
The authors have satisfactorily addressed my comments on the earlier version of the manuscript and I may recommend the manuscript for publication.